# Metagenomic data from cerebrospinal fluid permits tracing the origin and spread of *Neisseria meningitidis* CC4821 in China

Hongbin Chen [1,4✉], Mei Li[2,4], Shangyu Tu[3], Xiaoyang Zhang[1], Xiaojuan Wang[1], Yawei Zhang[1], Chunjiang Zhao[1], Yinghui Guo[2✉] & Hui Wang [1✉]

Metagenomic next-generation sequencing (mNGS) is useful for difficult to cultivate pathogens. Here, we use cerebrospinal fluid mNGS to diagnose invasive meningococcal disease. The complete genome sequences of *Neisseria meningitidis* were assembled using *N. meningitidis* of ST4821-serotype C isolated from four patients. To investigate the phylogeny, 165 CC4821 *N. meningitidis* genomes from 1972 to 2017 were also included. The core genome accumulated variation at a rate of $4.84 \times 10^{-8}$ substitutions/nucleotide site/year. CC4821 differentiated into four sub-lineages during evolution (A, B, C, and D). While evolving from sub-lineage A (early stage) to sub-lineage D (late stage), the ST and CC4821 serotype converged into the ST4821-serotype C clone. Most strains of sub-lineage D were isolated from invasive meningococcal disease, with increasing resistance to quinolones. Phylogeographic analysis suggests that CC4821 has spread across 14 countries. Thus, the selective pressure of quinolones may cause CC4821 to converge evolutionarily, making it more invasive and facilitating its spread.

[1] Department of Clinical Laboratory, Peking University People's Hospital, Beijing, China. [2] Department of Clinical Laboratory, Children's Hospital of Hebei Province, Shijiazhuang, Hebei, China. [3] Department of Clinical Medicine, Peking University People's Hospital, Peking University Health Science Center, Beijing, China. [4] These authors contributed equally: Hongbin Chen, Mei Li. ✉email: chenhongbin_pkuph@163.com; HeBeiGyh2021@126.com; whuibj@163.com

Meningitis is a serious life-threatening inflammation of the meninges and subarachnoid space that occurs on a global scale in individuals of all ages, causing high rates of morbidity and mortality. *Neisseria meningitides* are one of the important pathogens causing bacterial meningitis. The upper respiratory tract is the only reservoir of *N. meningitidis*[1], although some individuals develop invasive diseases. Once this bacterium invades the blood, human bactericidal antibodies, complement, and phagocytes are activated to eliminate the bacteria. However, if the host defense system is unable to clear the bacteria, this infection develops into bacteremia, followed by invasion into the meninges, which then quickly develops into meningitis and pathogenic sepsis[2]. Typical signs of meningitis include a stiff neck, fever, and changes in mental status. Susceptible populations to *N. meningitidis* infection include infants, adolescents, and young individuals (under the age of one)[3,4]. In addition, individuals with complement system defects and asplenia are also at high risk[4]. Meningococcal meningitis has a mortality rate between 5% and 15%, which can rise to 50% if left untreated[5,6]. Over one-third of individuals with the invasive meningococcal disease survive but experience major complications, with 9% suffering from long-lasting sequelae[7,8].

Abnormal results of general laboratory tests for meningococcal meningitis cerebrospinal fluid include high opening pressure, increased white blood cell count, increased protein concentration, and decreased glucose concentration. Clinicians can make preliminary clinical diagnoses of meningococcal meningitis based on the symptoms and abnormal laboratory test results to promptly treat affected patients. However, to make a precise pathogenic diagnosis, it is necessary to isolate or detect *N. meningitidis* in the patient's cerebrospinal fluid, blood, or skin lesions. Pathogenic detection methods for *N. meningitidis* include Gram staining, culture, and polymerase chain reaction (PCR). The majority of meningococcal meningitis patients (85%) are diagnosed using Gram staining[9]. In meningococcal meningitis, the positive rate of *N. meningitidis* cultured in blood culture is approximately 50%, while that in the cerebrospinal fluid is 75%[9]. Compared with the culture method, PCR can be used for faster detection of *N. meningitidis* and non-viable bacteria[10]. Loop-mediated isothermal amplification can be used to detect *N. meningitidis* more accurately than culture and PCR[11,12]. In recent years, metagenomic next-generation sequencing (mNGS) has started to be applied in the detection of infectious disease pathogens[13–16]. In the present study, mNGS was used to successfully detect *N. meningitidis* in the cerebrospinal fluid of four patients with suspected meningococcal meningitis, and the genome of *N. meningitidis* was assembled for further research.

The incidence of invasive meningococcal disease has been declining since the early 2000s[17]. In 2017, the incidence of invasive meningococcal disease in Europe was 0.45 to 1.33 cases per 100,000 population, with a case fatality rate of 9.7%[18]. Meningococcal B (MenB) (51%) is currently the most common serotype in Europe, followed by MenW (17%) and MenC (16%)[18]. In the United States, invasive meningococcal disease has declined sharply in recent decades, and by 2017, the incidence was only 0.13/100,000[18]. The distribution of serotypes from 2006 to 2015 was as follows: MenB, 38%; MenY, 30%; MenC, 24%; and MenW, 7%[18]. In Central and South America, the current incidence of invasive meningococcal disease is less than 1 in 100,000, with MenC being the most common serotype[18]. Historically, the meningitis belt in North Africa has had the highest incidence of invasive meningococcal disease in the world. MenA used to be the most common serotype in sub-Saharan Africa, giving rise to several outbreaks of invasive meningococcal disease. Since the large-scale vaccination of individuals with MenA conjugate vaccine MenAfriVac in 2010, the incidence of invasive

meningococcal disease has fallen rapidly, resulting in a shift of the epidemiology of the invasive meningococcal disease. The current most common serotypes are mainly MenC, MenW, and MenX[18]. In Australia, the incidence of the invasive meningococcal disease dropped after MenC conjugate vaccine vaccination, but gradually increased (1.5/100,000) from 2013 to 2017[18]. Currently, MenW is the most common serotype[18]. In New Zealand, MenB is the most common serotype of invasive meningococcal disease[18]. After a MenB strain-specific OMV vaccine (MeNZBTM) inoculation, the incidence of invasive meningococcal disease dropped from 17.4/100,000 to 1.2/100,000[19]. Before the 1980s, the meningococcal disease was a very serious public health problem in China, reaching a peak of 403/100,000 in 1967, with MenA being the dominant serotype[20]. However, since the widespread vaccination against MenA was started in the early 1980s, the rates of meningococcal disease have dropped markedly and are currently below 1/100,000. Since then, MenC belonging to the cloning complex (ST4821) has become the most dominant serotype[21]. In this study, the *N. meningitidis* detected in the cerebrospinal fluid of four pediatric patients with mNGS also belonged to ST4821-MenC. We downloaded all the genome sequences of *N. meningitidis* belonging to CC4821 from http://pubmlst.org/neisseria[22], performed phylogenetic analysis, analyzed its evolutionary rate, traced its origin time, and investigated its evolutionary trajectory in the world.

## Results

**Epidemiology of CC4821**. The CC4821 clone of *N. meningitidis* was mainly prevalent in China (127/169, 75.1%), followed by the UK (20/169, 11.8%), with sporadic cases in Brazil, Colombia, Greece, India, Ireland, Italy, Japan, Malta, New Zealand, Spain, Sweden, and the USA (Fig. 1). Serotype B (92/169, 54.4%) was the most prevalent serotype of CC4821, followed by serotype C (65/169, 38.5%) and serotype W (10/169, 5.9%). The CC4821 clones included in this study had 49 ST types, of which ST4821 (45/169, 26.6%) was the dominant ST type, followed by ST3200 (31/169, 18.3%), ST5664 (14/169, 8.3%), ST8491 (9/169, 5.3%), ST3436 (7/169, 4.1%), ST5798 (6/169, 3.6%), and ST9454 (5/169, 3.0%). Among 45 ST4821s, serotype C (38/45, 84.4%) was the most prevalent serotype, followed by serotype B (7/45, 15.6%). ST3200 was the opposite of ST4821, with serotype B (29/31, 93.5%) being the dominant serotype, while serotype C accounted for only 6.5% (2/31). Four ST5798s were from Shanghai, all of which were serotype B. The geographical distribution of ST3200 was as follows: UK (16 serotype B, 1 serotype C), China (7 serotype B, 1 serotype C), Brazil (2 serotype B), Ireland (1 serotype B), Italy (1 serotype B), Greece (1 serotype B), and Malta (1 serotype B).

**Phylogenetic reconstruction of the CC4821 populations**. After the recombination regions were excluded, a total of 169 core genomes were used to reconstruct the evolutionary history of *N. meningitidis* CC4821 (Fig. 2 and Supplementary Figure 3). These data revealed that the core genome accumulated variation at a rate of $4.8 \times 10^{-8}$ substitutions per nucleotide site per year [95% highest probability density (HPD), $4.2 \times 10^{-8}$ to $5.7 \times 10^{-8}$]. Phylogeny analysis showed that CC4821 differentiated into four sub-lineages during evolution: sub-lineage A, B, C, and D (Fig. 2). These four sub-lineages shared the most recent common ancestor (tMRCA), dating back to approximately 1941 (Fig. 2; 95% HPD 1924~1954). From sub-lineage A (early stage) to sub-lineage D (late stage), the serotype of CC4821 converged, mainly into the serotype C clone (Fig. 2). A cluster of serotype W isolates emerged in sub-lineage C (Fig. 2). In the CC4821 phylogenetic tree, isolates from different locations were found to be intermingled (Fig. 2), indicating that frequent transmission had

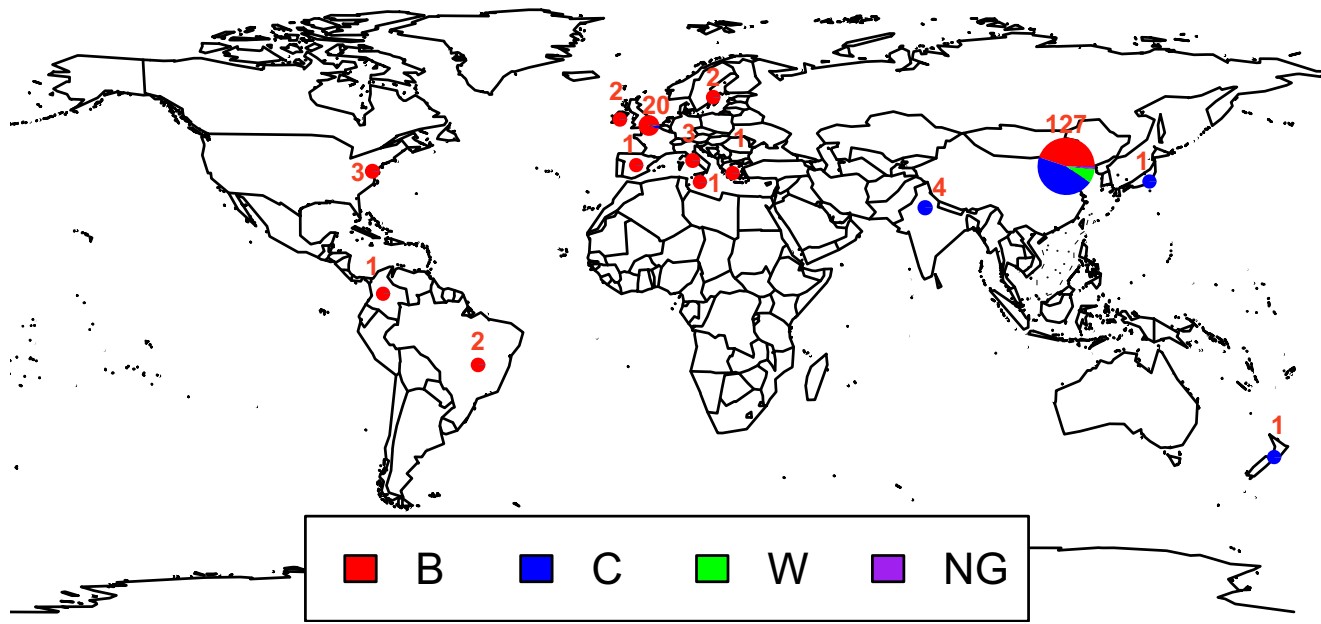

**Fig. 1 The global distribution of isolates for this study.** A total of 169 *N. meningitidis* isolates were sampled from China, the UK, Brazil, Colombia, Greece, India, Ireland, Italy, Japan, Malta, New Zealand, Spain, Sweden, and the USA.

occurred between different locations. In sub-lineage A, more strains were isolated from carriers than invasive diseases, while all other strains were isolated from invasive disease, except for nine strains in sub-lineage D (Fig. 2).

**Phylogeographic reconstruction of CC4821 populations in the world.** The worldwide spread of CC4821 was reconstructed using Bayesian phylogeography. As a result, CC4821 was found to spread to 14 countries (Fig. 3). This cross-country and cross-continent spread may be related to globalization.

**The evolutionary dynamic of the accessory genome of CC4821 populations.** We extracted the accessory genomes of 169 *N. meningitidis* CC4821 isolates and found that the accessory genomes were quite different (Supplementary Figure 4). This suggests that CC4821 underwent some changes in its accessory genome during its evolution, which is worthy of attention.

During the evolution of CC4821, antimicrobial resistance did not change. The vast majority of CC4821 carried *farB* and had 23 S *rRNA* mutation. Several isolates carried *farA*, *mtrD*, *tet(B)*, *tetR*, and *rpsJ* and had penicillin-binding protein (PBP) and *gyrA* mutations. *FarA* and *farB* are components of the FarAB efflux pump, which are related to the resistance of antibacterial free fatty acids. The 23 S *rRNA* mutation confers resistance to macrolide antibiotics. As a component of MtrCDE efflux complex, *mtrD* is related to macrolide resistance. *Tet(B)* and *tetR* are the tetracycline efflux proteins expressed in many Gram-negative bacteria. They confer resistance to tetracycline, doxycycline, and minocycline, but not tigecycline. The *rpsJ* gene encodes tetracycline-resistant ribosomal protection protein, which is associated with tetracycline. The PBP2 mutation is related to β-lactam resistance, while gyrase (encoded by *gyrA*), the target site of quinolones, mutated, giving rise to quinolone resistance. During the evolution from early sub-lineage A to late sub-lineage D, the resistance to quinolones increased quickly, and most sub-lineage D strains carried the amino acid change (T91I) in *gyrA*, which confers quinolone resistance (Fig. 4).

Over time, the virulence genes carried by CC4821 were found to not have changed much. Most isolates carried several virulence

genes, which are related to adherence, immune evasion, immune modulator, invasion, iron uptake, protease, and stress adaptation (Fig. 5). Moreover, no differences were observed in virulence genes between carrier strains and invasive strains, the reason being that some virulence genes were located on plasmids, which were not sequenced.

## Discussion

At present, the immunization schedule for children with meningococcal disease in China is as follows: children between 6–18 months of age receive two doses of serotype A capsular polysaccharide vaccine, followed by one dose of serotype A plus C capsular polysaccharide vaccine at 3 and 6 years of age. Although China has generally implemented meningococcal disease immunizations in children, sporadic cases of meningococcal disease are still being reported. Meningococcal meningitis is more common in children, and the four cases evaluated in this study were in patients between the ages of 9 and 13 years. The isolation of *N. meningitidis* from the cerebrospinal fluid is the most direct evidence for the diagnosis of meningococcal meningitis. However, none of the four cases in this study were cultured with *N. meningitidis*. As such, mNGS was used to confirm the etiological diagnosis, although only in four cases, indicating that the sensitivity of mNGS may be higher than that of traditional culture methods in the etiological diagnosis of central nervous system infections. mNGS has been used in the diagnosis of central nervous system infections[13–16]. When meningococcal meningitis is clinically suspected but the culture is negative, mNGS can be used to confirm the etiological diagnosis in as little as 24 h in order to obtain early etiological evidence and enable rapid treatment.

There were five epidemics of meningococcal serotype A in China between 1938 and 1977, during which the incidence reached a peak of 403/100,000 in 1967[20]. However, since China began to vaccinate using the MenA capsular polysaccharide vaccine in the 1980s, the incidence of MenA has fallen markedly. Subsequently, the expansion of the highly virulent clone complex (CC) 4821 led to a change in the epidemiology of meningococcal disease, from MenA to MenC[23]. Previous surveillance data showed that MenC CC4821 had an outbreak in China between

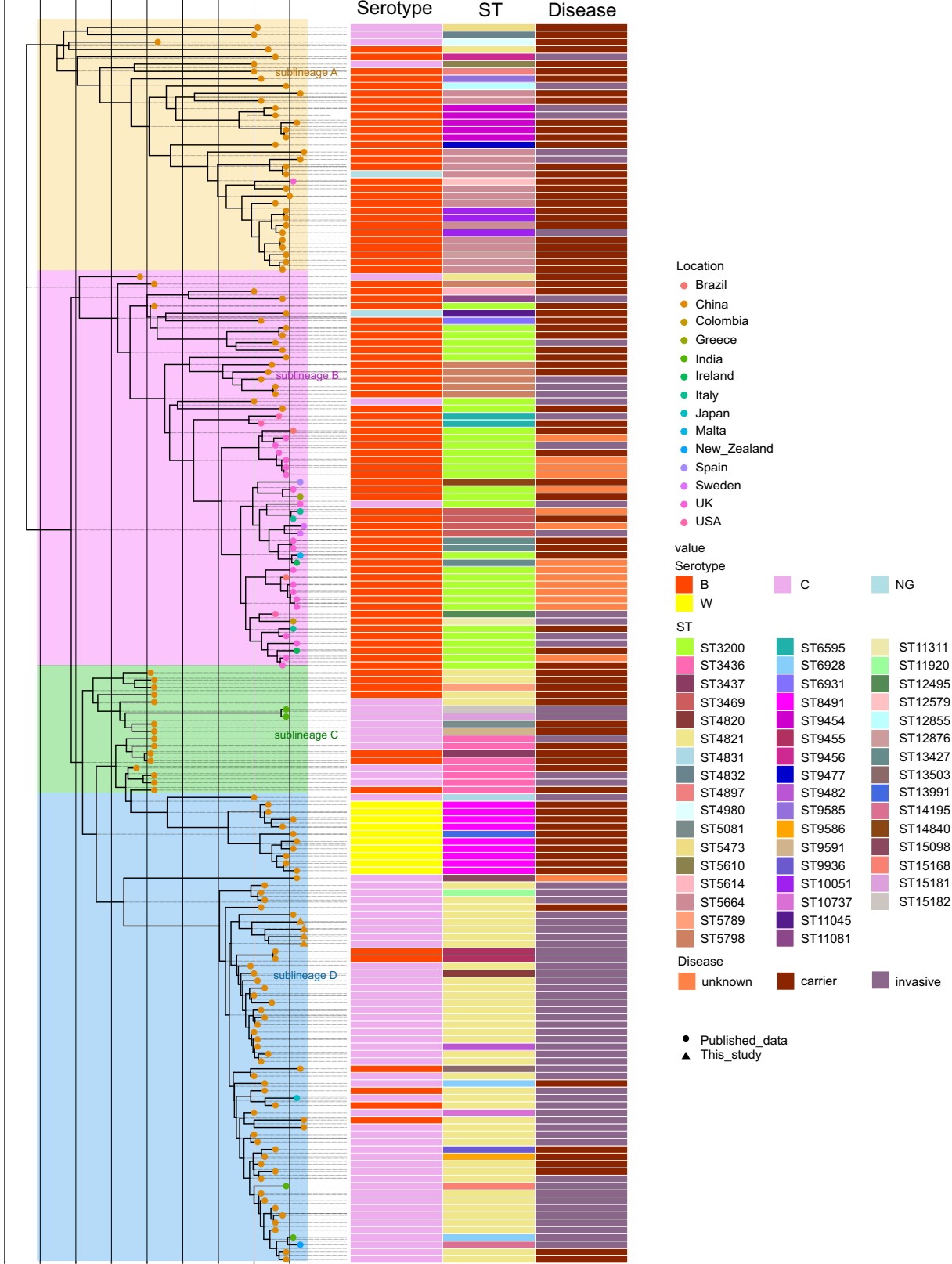

**Fig. 2 Maximum clade credibility (MCC) tree and distributions of serotype, sequence type (ST), and disease type amongst CC4821 isolates.** MCC tree of CC4821 population based on BEAST analysis. The tips of the tree were colored according to the location of which isolates were sampled. The time scale was shown below the tree. The four colored boxes in the MCC tree represented the four lineages of CC4821, sub-lineages A, B, C, and D. The distributions of serotype, sequence type (ST), and disease types were plotted against the core genome phylogeny.

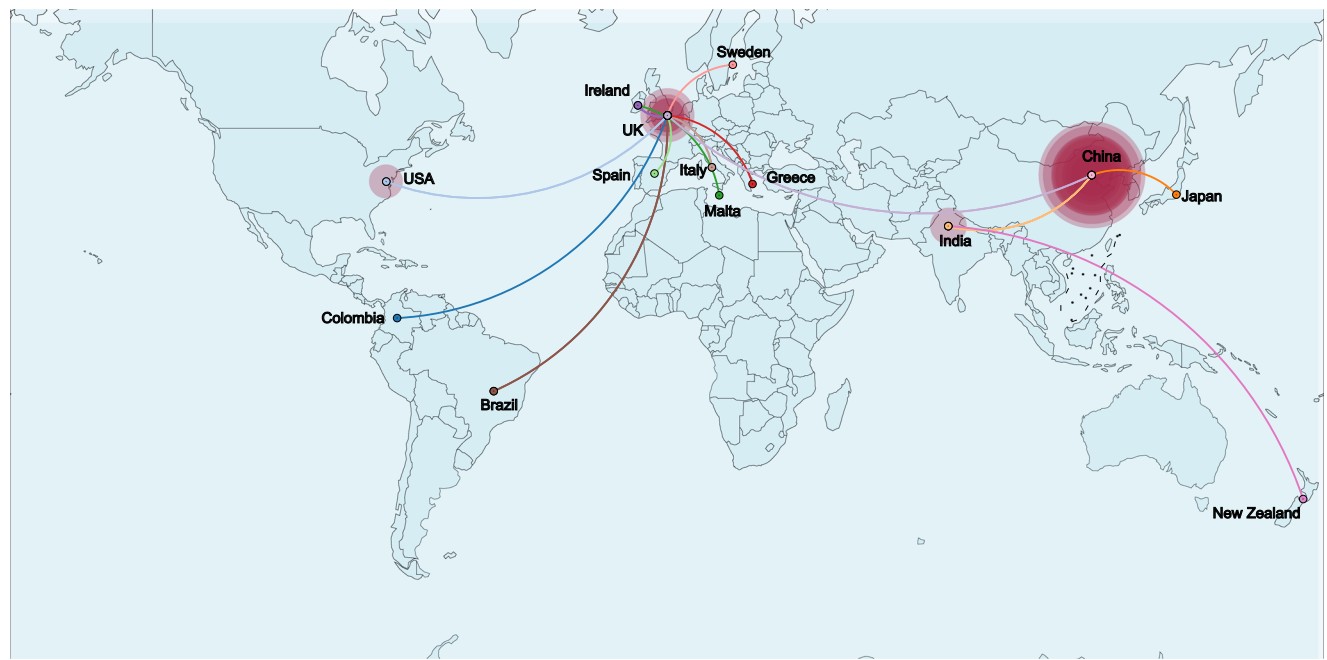

**Fig. 3 Bayesian reconstruction of the spread of CC4821 in the world.** A discrete diffusion model was used to reconstruct the finer-scale geographical dispersal of CC4821 in the world. Point colors represented the sampling countries, and line colors represented the destination countries. Eastward movements are depicted by lines with an upward curvature, while westward movements are depicted by lines with a downward curvature.

2003 and 2005[24], and the MenC strain has been the dominant clone between 2005 and 2012[25]. The *N. meningitidis* strains of the four pediatric cases evaluated in this study were all ST4821, belonging to the CC4821 clone complex. Therefore, we conducted further studies on the phylogeny of *N. meningitidis* CC4821. This study also suggests that CC4821 has undergone a process of converging evolution, from the early polyclonal to the late single clone ST4821-MenC, which indicates that the outbreak of CC4821 is caused by the expansion of the MenC clone.

Also worth noting is the evolution of CC4821, from the early carrier and invasion strains coexisting (sub-lineage A) to the recent tendency to be more prone to invasion (sub-lineage D). This phenomenon has also been found in other clones. Compared with the *N. meningitidis* ST-41/44 CC isolated between 1971-2000, the strains isolated between 1988-2015 are more prone to invasive disease[26]. However, the mechanism underlying the tendency of CC4821 to be more likely to occur in invasive diseases during evolution requires further study.

In this study, the number of quinolone-resistant strains in CC4821 gradually increased over time, and most of the strains in the late lineage (sub-lineage D) were resistant to quinolones. Quinolone resistance in *N. meningitidis* in China has evolved over time, from the era of no quinolone resistance (before ≈1985) to the era of quinolone resistance (none versus >70%), especially in the highly virulent CC4821 and CC5 lineages[24]. In this study, the results of phylogenetic analysis suggest that quinolone resistance plays an important role in the evolution of CC4821[23], and fluoroquinolone selective pressure is the main reason for the outbreak of CC4821 in China. A recent study has shown that over half of quinolone-resistant *N. meningitidis* isolates obtained quinolone resistance genes from three commensal *Neisseria* species, *N. lactamica*, *N. cinerea*, and *N. subflava*, through horizontal gene transfer[27]. Nonetheless, in the present study, quinolone resistance was found to be caused by a target mutation, which was related to the selective pressure of quinolones.

To the best of our knowledge, this study is the first to deduce the evolutionary rate and transmission of CC4821. Recently, CC4821 has spread to new countries, including France

(2009–2011)[28], the UK (2011-2014)[28], USA (2007–2016)[28], Canada (2013–2015)[29], Japan (2017)[30], and India (2017)[31]. This reveals that the geographical spread of CC4821 is increasing, which is worthy of continued attention.

## Methods

**Patients and samples.** Patient 1: A 9-year-old previously healthy boy presented with fever and confusion of half a day to the Children's Hospital of Hebei Province. Upon examination, the patient was found to be febrile (36.7 °C), with a pulse of 120 beats per minute and a respiration rate of 38 breaths per minute. Laboratory examination revealed peripheral leukocytosis with an increased neutrophil count and elevated C-reactive protein levels. Lumbar punctures revealed increased pressure in the cerebrospinal fluid. Elevated cell counts and protein levels but lower glucose levels were also observed. A cerebrospinal fluid smear was positive for Gram-negative diplococcus. *N. meningitidis* was identified using mNGS in a cerebrospinal fluid sample. Upon detection, the patient was administered a treatment of meropenem and 15 days of penicillin, and he was discharged from the hospital after his complete recovery.

Patient 2: A 13-year-old boy presented to the Children's Hospital of Hebei Province with a two-day history of fever and a one-day history of headache and lethargy. Upon examination, the patient was found to have a body temperature of 36.7 °C, with a pulse of 128 beats per minute and a respiration rate of 22 breaths per minute. A lumbar puncture was performed on day 2 of hospitalization after the administration of revealed pleocytosis (199858 white blood cell $\mu L^{-1}$), 3.95 g $L^{-1}$ protein content, and 0.16 mmol $L^{-1}$ glucose content. Although this patient's cerebrospinal fluid and blood cultures were negative, he was diagnosed with invasive meningococcal disease by mNGS. He was then treated with meropenem and penicillin for two weeks and recovered.

Patient 3: A 13-year-old boy was admitted to the Children's Hospital of Hebei Province with a two-day history of fever, headache, emesis, and confusion. Upon examination, the patient was found to have a temperature of 36.4 °C, with a pulse of 72 beats per minute, a blood pressure of 107/67 mmHg, and a respiration rate of 20 breaths per minute. The first laboratory test indicated hematologic abnormalities of the white blood cell ($18.7 \times 10^9 L^{-1}$) and polymorphonuclear neutrophils (87.0%). The results showed cerebrospinal fluid pleocytosis (59440 white blood cell $\mu L^{-1}$), elevated protein content (6.1 g $L^{-1}$), and decreased glucose content (0.03 mmol $L^{-1}$). Although the results of infectious disease cultures were negative, *N. meningitidis* was detected by mNGS in his cerebrospinal fluid. The patient was then treated with chloramphenicol and penicillin for two weeks, but was discharged from the hospital with incomplete recovery.

Patient 4: A 12-year-old girl presented to the Children's Hospital of Hebei Province with a two-day history of fever and headache, a one-day history of emesis, and a half-day history of confusion. Upon examination, she was found to be febrile (37.5 °C), with a pulse of 104 beats per minute, a blood pressure of 111/64 mmHg, and a respiration rate of 28 breaths per minute. The cerebrospinal fluid evaluation

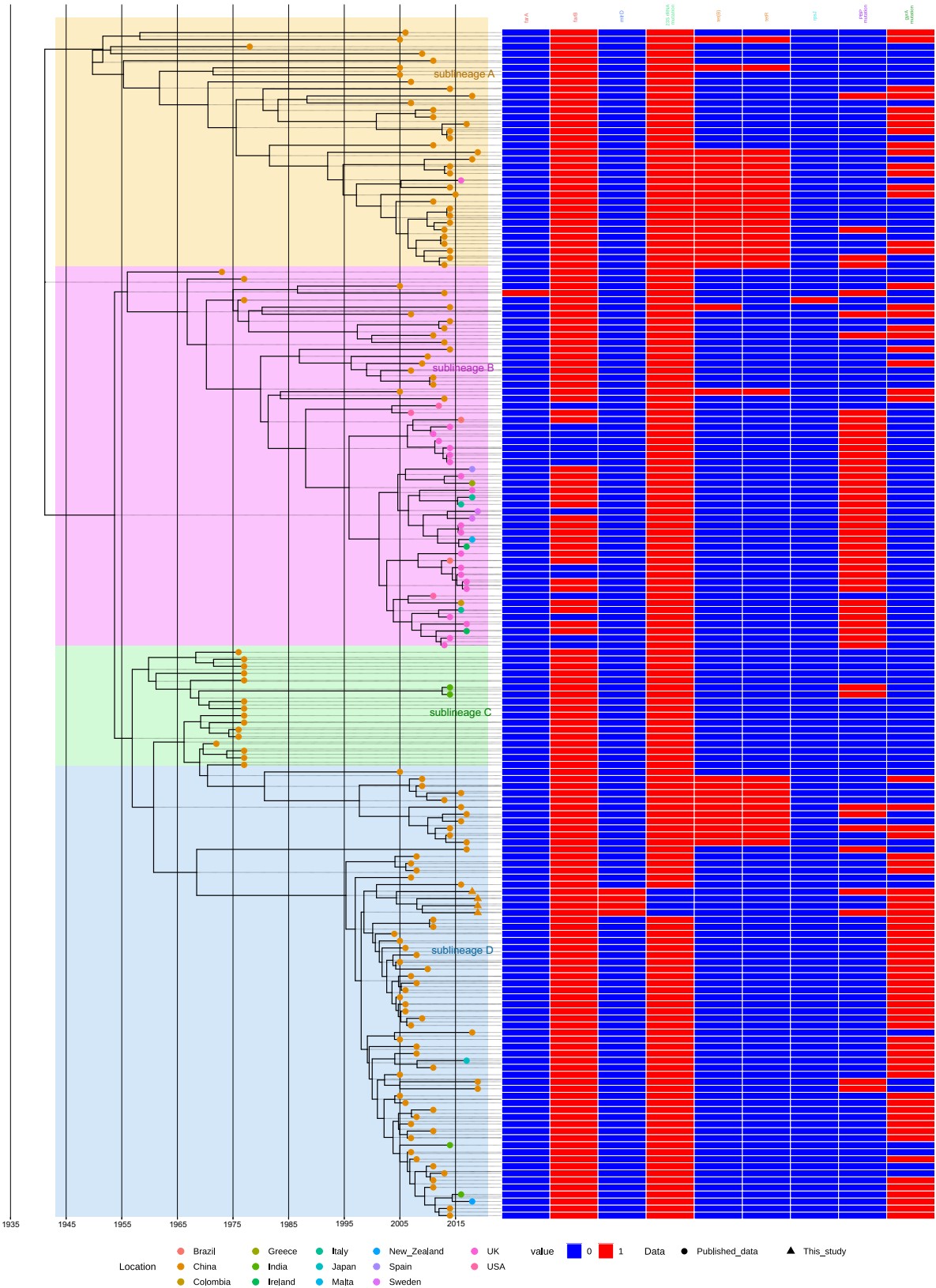

**Fig. 4 Presence or absence of antimicrobial resistance (AMR) genes amongst CC4821 isolates.** The distribution of AMR genes was plotted against the core genome phylogeny. The names of different types of AMR genes were displayed in different colors.

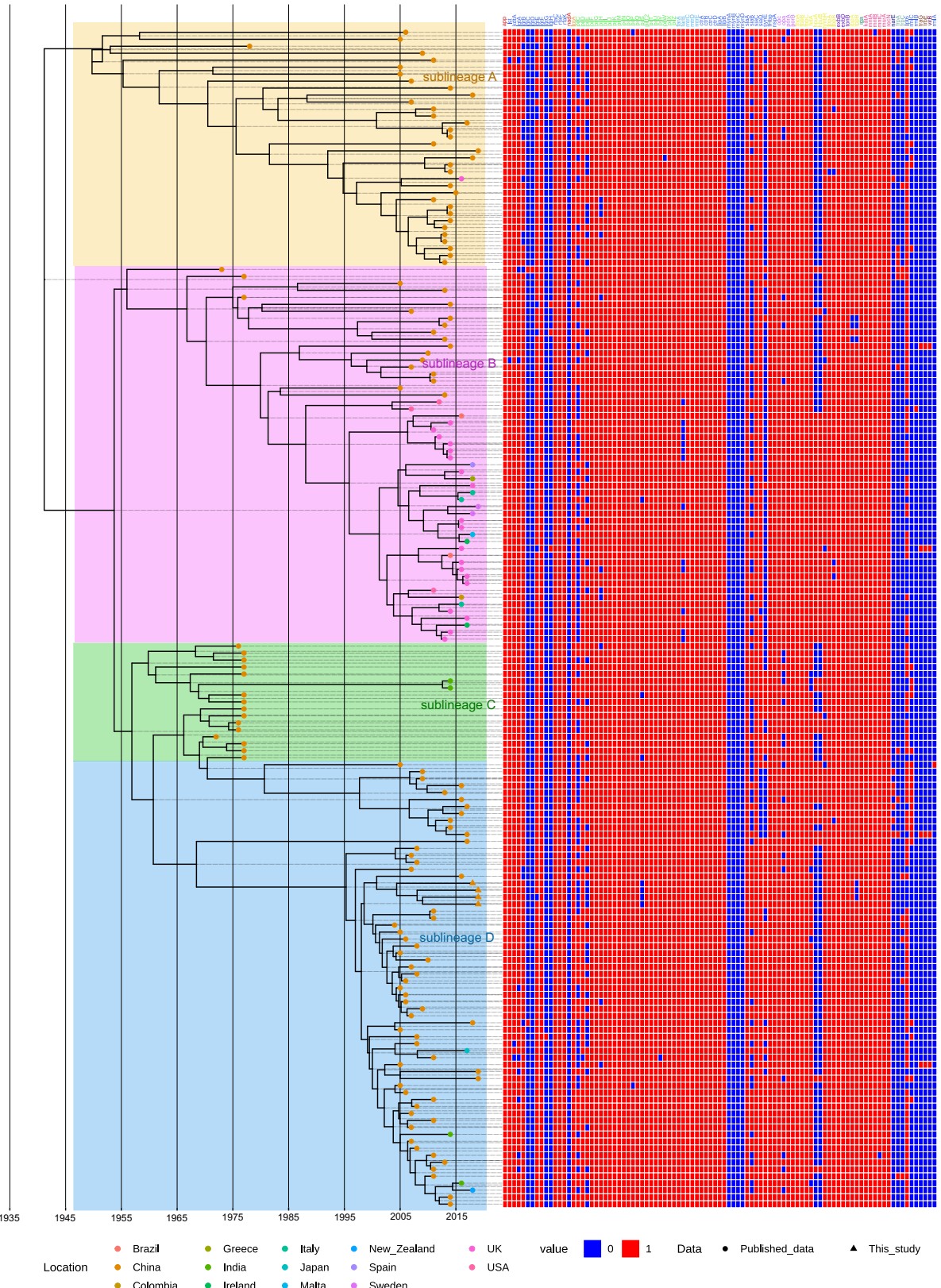

**Fig. 5 Presence or absence of virulence genes amongst CC4821 isolates.** The distribution of virulence genes was plotted against the core genome phylogeny. The names of different types of virulence genes were displayed in different colors.

revealed cerebrospinal fluid pleocytosis (19858 white blood cell μL$^{-1}$), protein (3.95 g L$^{-1}$), and glucose (0.16 mmol L$^{-1}$). Although the results of infectious disease evaluation were negative, *N. meningitidis* was detected by mNGS in a cerebrospinal fluid sample. The patient was then administered a course of meropenem, and her symptoms resolved before discharge.

*N. meningitidis* identified by mNGS in the cerebrospinal fluid specimens of these four patients all belonged to ST4821-serotype C. To investigate the phylogeny of CC4821, we also included the genome of 165 CC4821 *N. meningitidis* from the PubMLST Neisseria genome database (http://pubmlst.org/neisseria)[22]. Among the 169 *N. meningitidis* isolates, 127 were from China, 20 were from the UK, four were from India, three were from Italy, three were from USA, two were from Brazil, two were from Ireland, one was from Colombia, one was from Greece, one was from Japan, one was from Malta, one was from New Zealand, one was from Spain, and one was from Sweden (Fig. 1). This study has included all the sequenced CC4821 genomes downloaded from the PubMLST Neisseria genome database (http://pubmlst.org/neisseria)[22] (Supplementary Data 1).

**Metagenomic DNA-seq**. DNA was extracted from the cerebrospinal fluid using QIAamp DNA Microbiome Kit (Qiagen). Paired-end libraries were prepared with the Vazyme TruePrepTM DNA Library Prep Kit (Vazyme). Sequencing was performed using the NextSeq 550 System (Illumina), resulting in paired reads of 150 bp in length. Taxonomy was evaluated using Kraken2 v2.0.8beta -db $database -threads 6 -unclassified-out f.unclassified -classified-out f.classified -report f.classified.0.report -use-names -gzip-compressed -confidence 0.8 file.fastq > output_file[32]. Low-quality bases were excluded from the raw mNGS sequencing data before aligning them to the reference genome MC58 using BWA[33]. Then, the consensus program of Samtools[34] was used to obtain the genome sequence according to the aligned BAM results. During consensus, the parameter -min-MQ 20 was selected to obtain a more accurate sequence. The resultant assemblies were annotated using Prokka v1.12 -mincontiglen 100 -prefix $sample $sample.fasta -force[35]. Multilocus sequence types were assigned using PubMLST (https://pubmlst.org)[22]. The genomes of 169 *N. meningitidis* isolates were scanned against a list of genes in regions A (capsule synthesis), B (capsule translocation), C (capsule transport), and E of the cps locus for each serogroup. The serotype was determined based on the identification of at least one serogroup-specific gene[36].

**Preliminary phylogenetic analysis of CC4821**. The core genome (1114 core genes and 169 isolates) was extracted using Roary v3.11.2 -e --mafft -p 4 *.gff -f roary_output[37]. The recombination regions were then removed from the core genome using ClonalFrameML RAxML_file core_gene_alignment.fasta output_file_name before conducting phylogenetic analysis[38]. The maximum likelihood phylogenetic tree was constructed in RAxML v8.2.10 using a GTR model and 1000 bootstrap replicates[39]. For rooting, the chromosome of *N. meningitidis* isolates MC58 (GenBank accession no. NC_003112) was used, as it represented the most closely related outgroup.

**Phylogeographic analysis of CC4821**. A timed phylogeny was estimated from the core genome alignment of CC4821 using BEAST v2.4.7[40]. The TN93 substitution model was selected based on evaluation of all possible substitution models in bModelTest. Then, all combinations of strict, relaxed lognormal, relaxed exponential clock models and skyline exponential population models were evaluated. For each of these, three independent chains were run for 200,000,000 iterations sampling every 20,000 steps, and convergence was checked by inspecting the effective sample sizes (ESS) and parameter value traces in Tracer v1.6.0. The best-supported model was an exponential population model with a strict clock (Supplementary Figure 1 and Supplementary Figure 2). Next, the geographic origin of the entire dataset based on the.root BEAST output file was determined. The maximum clade credibility tree was generated in TreeAnnotator and plotted in ggtree[41]. Finally, the phylogeographic spread was visualized by SPREAD3[42].

**Accessory genome analysis of CC4821**. The accessory genome was extracted using Roary v3.11.2[37]. The antimicrobial resistance and virulence genes were identified using CARD[43] and VFDB[44], respectively, with a minimum query coverage value of 80% and a similarity threshold value of 90%.

**Statistics and reproducibility**. The maximum likelihood phylogenetic tree was constructed in RAxML v8.2.10. Timed phylogeny was estimated using BEAST v2.4.7. Statistical analyses were performed using R (ver. 4.0.5). Figures were drawn using the ggplot2 package in R (ver. 3.3.5).

**Reporting summary**. Further information on research design is available in the Nature Research Reporting Summary linked to this article.

## Data availability
The sequence data of the clinical *N. meningitidis* isolates were deposited under BioProject ID PRJNA722594. Other relevant data that support the findings of this study are available in the PubMLST Neisseria genome database (https://doi.org/10.12688/wellcomeopenres.14826.1) (ref. [22]).

## Code availability
The software used is described in the Methods section. The open source software includes the following external tools: Kraken2 v2.0.8beta, Prokka v1.12, Roary v3.11.2, RAxML v8.2.10, and BEAST v2.4.7. Descriptions of all open source code and software are included in the Methods section, and further details are available on request.

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

## Acknowledgements

This study was supported by the funding of the National Science and Technology Major Project (2018ZX10305410), Beijing Municipal Science and Technology Commission program (Z191100006619100), and national key research and development program of China (2018YFE0102100).

## Author contributions

H.W. and H.C. designed and supervised the study. M.L. and Y.G. obtained the samples and clinical details. S.T., X.Z., X.W., Y.Z., and C.Z. assisted in extracting the DNA and performing the mNGS. H.C. performed the data analysis. H.C. and M.L. wrote the manuscript. H.C. was a major contributor to writing the manuscript. All authors read, edited, and approved the final manuscript.

## Ethics approval and consent to participate

This study was approved by the research ethics board at Peking University People's Hospital (2019PHB134).

## Competing interests

The authors declare no competing interests.
