## [Peer Review File · Communications Biology]

Reviewers' comments:

REVIEWER #1:

1. Brief summary of the manuscript

This manuscript describes metagenomic next generation sequencing (mNGS) of four *N. meningitidis* strains from four patients with invasive meningococcal disease, and how mNGS was used to confirm the diagnosis of the patients, when a cultured sample could not be obtained. It furthermore describes the comparison of the four strains to 37 previously whole genome sequenced *N. meningitidis* strains, to assess the phylogeny of bacterial strains within this clonal complex, and to assess the origin of the now prevalent Meningitis C strain which is caused by strains in the CC4821 clonal complex. The antibiotic resistance and the virulence genes of each of the 41 strains are found and compared to the phylogeny and the timed origin of the strains, the same is done for virulence genes. It is discussed whether quinolone usage and the resistance to this drug, is driving the evolution of MenC CC4821 strains and their prevalence in outbreaks.

2. Overall impression of the work

Using mNGS to sequence *N. meningitidis* from the cerebrospinal fluid and using this to diagnose the patients is an excellent use of sequencing. I find this study to be very interesting, and to the best of my knowledge, it is novel to assess the origin of the *N. meningitidis* clonal complex 4821. I think that one could highlight the quinolone resistance mutation in *gyrA* even more, as this is a strong supporter of the phylogenetic analysis, and the following results of the BEAST analysis.

However, some of the bioinformatic methods and tools used in the paper are only superficially described. Adding what parameters and settings used for each method would make the reproducibility much higher. If the default parameters were used, this should be stated, however most of these tools needs specifications. Furthermore, it would also make the results more transparent, if we were shown more of the results, this could be statistical analysis and output from the tools, submitted as supplementary figures, or integrated in the results. In the following specific comments, I have written where more results/parameters could be added. But I will highlight here, that adding the bootstrap values to the phylogenetic tree from RAXML, or to mention them in the text will give an added credibility, as the rest of the bioinformatic part relies heavily on this tree. The values could be added to Figure S3, or just adding the raw file of the best tree from RAXML, with bootstrap values, as a supplementary text file.

3. Specific comments

Lines 57-59: Could be changed to infants and adolescents. And I would change "In addition, complement system defects and asplenia are also at high risk crowd" to "In addition, persons with complement system defects and asplenia are also at high risk"

Line 74: Please add what LAMP is an abbreviation of.

Lines 75-76: A reference to this claim would be suitable. Not that it is not believable, it would just be more correct to site a few papers here. The references are there in the discussion on line 270.

Line 104: Can you specify what is meant by "almost all the genome sequences of CC4821 *N. meningitidis*"?

Line 155: Pubmlst should be cited by this paper <https://pubmed.ncbi.nlm.nih.gov/30345391/>

Line 160: "This study has included as much as possible all the sequenced CC4821 genomes". Here you could describe how you chose the genomes you used.

Lines 165-172: Could you add the settings with which Kraken, SPAdes, Prokka, PubMLST, Roary and ClonalFrameML was run used?

Line 166: Could add the settings and parameters for SPAdes? If you used metaSPAdes to assemble the genome from metagenomic data, you should write that instead. You also never give us any numbers on how well the assembly goes in the result section.

Lines 210-211: Could you add the length of the core genome? Alternatively, the number of genes in the core genome. You could also start this section with an overview of how well the assembly of the 4 mNGS assembly went. It could be the length of the total assembly, the NG50 or one of their standard measures. This is especially important because the information on Roary's webpage says "Roary is not intended for meta-genomics or for comparing extremely diverse sets of genomes". So, the reader needs assurance that your genomes are assembled as well as an isolated WGS sample would be.

Figure 2: Could you add "Disease" next to "Serotype" and "ST"?

Lines 250-252: This part is very believable and really underlines the phylogeny. However, the sentence is hard to understand, and could use rephrasing. I would suggest changing the last part to something in the line of: "and all of sub-lineage B strains carries the amino acid change (T91I) in gyrA, which confer quinolone resistance"

Lines 253-255 and Figure 5. Could you comment on why some of the genomes not have any virulence genes? Are they faulty, are all of the virulence genes on a plasmid, which was not sequenced? How can it be invasive if it does not contain any virulence genes? It could be addressed here or in the discussion.

Lines 296-298: Do you mean "era of none quinolone" and "era of quinolone"? These lines are difficult to understand, and the message on them is very important.

Lines 301-303: In your study you find quinolone resistance due to a point mutation, how does that correspond to horizontal gene transfer? Could you discuss this a little further?

REVIEWER #2:

This article provides insight into the phylogeny and evolutionary rate of a CC4281 *N. meningitidis* since 1902 in China. This article is novel in that it looks at sequencing data and strain relatedness of strains found predominantly in China. The data set is from those publicly available as well as well as 4 sequences of cases in children that were obtained directly from CSF.

While the claims in the paper are related predominantly to what is circulating in China, as a reviewer I feel this paper will be impactful to those in the field. The data seems valid and reproducible.

As a reviewer I would be interested to see how these sequencing analysis relate to others worldwide. The authors elude to cases of CC4821 spreading to other distant regions over the past 10-15 years. If the sequence data is available for these cases, I would be interested to see how they have changed in terms of phylogeny and the evolution rate since spreading worldwide. Does the evolution rate remain the same as they spread from China or is it remaining consistent. And are they similar phylogenetically or becoming more diverse? Also are they gaining antimicrobial resistance as they spread worldwide. If these were available, I would like to see a bit of this information (the authors indicate most but all the publicly available data was included-I am wondering if these were the ones left out). I feel this would increase the impact of the paper if possible to do.

REVIEWER #3:

The mNGS is proved to be useful tool for diagnostic purpose in general and in the epidemiological

study of diseases. No prior knowledge of pathogen is required for mNGS. Since the technology is widely used in research and healthcare institutes, the technology will tremendously contribute to the understanding of disease epidemiology and transmission which eventually translate into treatment and prevention. Although whole genome sequencing is out of my expertise, but as a general reader this study presents well by transforming the complexity of whole genomic data into a rather simple map, graph, and phylogenetic tree. Moreover, the main/important findings are easy to understand what the author tries to convey to readers.

The key interesting point of this study is being able to retrieve a whole genome of *Neisseria meningitidis* directly from samples (without culture) and be able to depict the evolutionary history of the main serotype and subtype of *N. meningitidis* isolates in China using genetic information. The author provided sufficient introduction for the reader to understand the objective of his/her study. Other information such as material and methods are quite specific and would benefit scientists who work in the same research area. All data/information were included as expected. Discussion is sufficient for the proposed study. Conclusions and data interpretation are clearly described. The manuscript is suitable for publication after revision.

Minor;

The *N. meningitidis* should be italicized consistently throughout the manuscript.

Materials and methods should include the location of hospital where patients were admitted.

Line 41: Abstract is clear but the last sentence should be in the discussion not in the abstract "In conclusion, the selective pressure of quinolones caused CC4821 to converge evolution, which is more invasive and spreading"

Line 251-252. It appears there are 2 sentences here. Please revise the statement.

Line 267: should be 4 cases,..

Line 271-272: when patients clinically suspected of meningococcal meningitis but the culture is negative. mNGS can be used to confirmed the etiological diagnosis as soon as possible. Suggest the author should indicate "how long each test takes to deliver the result".

Line 298-300: need the reference for this statement " ...fluoroquinolone selective pressure is the main reason for the outbreak of CC4821 in China"

Line 302: should it be quinolone resistant "gene" from three commensal....

Line 306: Add a space between ...France (2009-2011) [39].

Line 336-338 and Supplemental Dataset 2: The spreadsheet shows that Probability value of Shanghai-China is 0.20199, while Beijing_China has only 0.06215. However in the file legend, it said "Beijing might be the root of CC4821 with over 20% probability". A little confusing here, please clarify.

Thank you for your positive assessment of our manuscript. Please find below a response to editorial enquiries raised.

Referee expertise:

Referee #1: Bioinformatics, comparative genomics

Referee #2: Neisseria and public health

Referee #3: Epidemiology

Reviewers' comments:

REVIEWER #1:

1. Brief summary of the manuscript

This manuscript describes metagenomic next generation sequencing (mNGS) of four *N. meningitidis* strains from four patients with invasive meningococcal disease, and how mNGS was used to confirm the diagnosis of the patients, when a cultured sample could not be obtained. It furthermore describes the comparison of the four strains to 37 previously whole genome sequenced *N. meningitidis* strains, to assess the phylogeny of bacterial strains within this clonal complex, and to assess the origin of the now prevalent Meningitis C strain which is caused by strains in the CC4821 clonal complex. The antibiotic resistance and the virulence genes of each of the 41 strains are found and compared to the phylogeny and the timed origin of the strains, the same is done for virulence genes. It is discussed whether quinolone usage and the resistance to this drug, is driving the evolution of MenC CC4821 strains and their prevalence in outbreaks.

2. Overall impression of the work

Using mNGS to sequence *N. meningitidis* from the cerebrospinal fluid and using this to diagnose the patients is an excellent use of sequencing. I find this study to be very interesting, and to the best of my knowledge, it is novel to assess the origin of the *N. meningitidis* clonal complex 4821. I think that one could highlight the quinolone resistance mutation in *gyrA* even more, as this is a strong supporter of the phylogenetic analysis, and the following results of the BEAST analysis.

However, some of the bioinformatic methods and tools used in the paper are only superficially described. Adding **what parameters and settings** used for each method would

make the reproducibility much higher. If the default parameters were used, this should be stated, however most of these tools needs specifications. Furthermore, it would also make the results more transparent, if we were shown more of the results, this could be statistical analysis and output from the tools, submitted as supplementary figures, or integrated in the results. In the following specific comments, I have written where more results/parameters could be added. But I will highlight here, that **adding the bootstrap values to the phylogenetic tree from RAxML**, or to mention them in the text will give an added credibility, as the rest of the bioinformatic part relies heavily on this tree. **The values could be added to Figure S3**, or just adding the raw file of the best tree from RAxML, with bootstrap values, as a supplementary text file.

3. Specific comments

Lines 57-59: Could be changed to infants and adolescents. And I would change “In addition, complement system defects and asplenia are also at high risk crowd” to “In addition, persons with complement system defects and asplenia are also at high risk”

Re: Thank you! We have modified as you suggested.

Line 74: Please add what LAMP is an abbreviation of.

Re: We have added the abbreviation of LAMP.

Lines 75-76: A reference to this claim would be suitable. Not that it is not believable, it would just be more correct to site a few papers here. The references are there in the discussion on line 270.

Re: Many thanks. We have cited a few papers here.

Line 104: Can you specify what is meant by “almost all the genome sequences of CC4821 N. meningitidis”?

Re: We have modified it as “we downloaded all the genome sequences of N. meningitidis belonging to CC4821 from <https://pubmlst.org/organisms/neisseria-spp>”.

Line 155: Pubmlst should be cited by this paper <https://pubmed.ncbi.nlm.nih.gov/30345391/>

Re: Thanks. We have cited this paper.

Line 160: “This study has included as much as possible all the sequenced CC4821 genomes”. Here you could describe how you chose the genomes you used.

Re: We have modified it as “This study has included all the sequenced CC4821 genomes downloaded from the PubMLST Neisseria genome database (<http://pubmlst.org/neisseria>)”.

Lines 165-172: Could you add the settings with which Kraken, SPAdes, Prokka, PubMLST, Roary, and ClonalFrameML was run used?

Re: For PubMLST, we use the web version. Settings for other softwares have been added.

Line 166: Could add the settings and parameters for SPAdes? If you used metaSPAdes to assemble the genome from metagenomic data, you should write that instead. You also never give us any numbers on how well the assembly goes in the result section.

Re: We used the following mapping-based approach for genome assembly: Low-quality bases were excluded from the raw mNGS sequencing data and then aligned to the reference genome MC58 using BWA. Then the consensus program of Samtools was used to obtain the genome sequence according to the aligned BAM results. During consensus, the parameter --min-MQ 20 was selected to get a more accurate sequence. We have added the detailed assembly method to the revised manuscript.

Lines 210-211: Could you add the length of the core genome? Alternatively, the number of genes in the core genome. You could also start this section with an overview of how well the assembly of the 4 mNGS assembly went. It could be the length of the total assembly, the NG50 or one of their standard measures. This is especially important because the information on Roarys webpage says “Roary is not intended for meta-genomics or for comparing extremely diverse sets of genomes”. So, the reader needs assurance that your genomes are assembled as well as an isolated WGS sample would be.

Re: We have added the number of genes in the core genome (1114 core genes, 169 isolates) in the manuscript.

Figure 2: Could you add “Disease” next to “Serotype” and “ST”?

Re: Thanks. We have added it.

Lines 250-252: This part is very believable and really underlines the phylogeny. However, the sentence is hard to understand, and could use rephrasing. I would suggest changing the last part to something in the line of: “and all of sub-lineage B strains carries the amino acid change (T91I) in gyrA, which confer quinolone resistance”

Re: Many thanks. We have modified this sentence according to your suggestion.

Lines 253-255 and Figure 5. Could you comment on why some of the genomes not have any virulence genes? Are they faulty, are all of the virulence genes on a plasmid, which was not sequenced? How can it be invasive if it does not contain any virulence genes? It could be addressed here or in the discussion.

Re: Yes, I think so. Several virulence genes were located on the plasmids, which were not sequenced. We have addressed this here.

Lines 296-298: Do you mean “era of none quinolone” and “era of quinolone”? These lines are difficult to understand, and the message on them is very important.

Re: Yes, You're right. We have corrected it.

Lines 301-303: In your study you find quinolone resistance due to a point mutation, how does that correspond to horizontal gene transfer? Could you discuss this a little further?

Re: Yes. In our study, we found quinolone resistance due to a point mutation. We have discussed this in the manuscript.

REVIEWER #2:

This article provides insight into the phylogeny and evolutionary rate of a CC4281 N. meningitidis since 1902 in China. This article is novel in that it looks at sequencing data and strain relatedness of strains found predominantly in China. The data set is from those publicly available as well as well as 4 sequences of cases in children that were obtained directly from CSF.

While the claims in the paper are related predominantly to what is circulating in China, as a reviewer I feel this paper will be impactful to those in the field. The data seems valid and reproducible.

As a reviewer I would be interested to see how these sequencing analysis relate to others worldwide. The authors elude to cases of CC4821 spreading to other distant regions over the past 10-15 years. If the sequence data is available for these cases, I would be interested to see how they have changed in terms of phylogeny and the evolution rate since spreading worldwide. Does the evolution rate remain the same as they spread from China or is it remaining consistent. And are they similar phylogenetically or becoming more diverse? Also are they gaining antimicrobial resistance as they spread worldwide. If these were available, I would like to see a bit of this information (the authors indicate most but all the publicly available data was included-I am wondering if these were the ones left out). I feel this would increase the impact of the paper if possible to do.

Re: Thank you very much for your affirmation of our study. According to your suggestion, we involved 169 isolates worldwide for phylogenetic analysis and antimicrobial resistance analysis.

REVIEWER #3:

The mNGS is proved to be useful tool for diagnostic purpose in general and in the epidemiological study of diseases. No prior knowledge of pathogen is required for mNGS. Since the technology is widely used in research and healthcare institutes, the technology will tremendously contribute to the understanding of disease epidemiology and transmission which eventually translate into treatment and prevention. Although whole genome sequencing is out of my expertise, but as a general reader this study presents well by transforming the complexity of whole genomic data into a rather simple map, graph, and phylogenetic tree. Moreover, the main/important findings are easy to understand what the author tries to convey to readers.

The key interesting point of this study is being able to retrieve a whole genome of Neisseria meningitidis directly from samples (without culture) and be able to depict the evolutionary history of the main serotype and subtype of N. meningitidis isolates in China using genetic information. The author provided sufficient introduction for the reader to understand the

objective of his/her study. Other information such as material and methods are quite specific and would benefit scientists who work in the same research area. All data/information were included as expected. Discussion is sufficient for the proposed study. Conclusions and data interpretation are clearly described. The manuscript is suitable for publication after revision.

Re: Thank you very much for your affirmation of our study.

Minor;

The *N. meningitidis* should be italicized consistently throughout the manuscript.

Re: Many thanks. The *N. meningitidis* have been italicized consistently throughout the manuscript.

Materials and methods should include the location of hospital where patients were admitted.

Re: Thank you for pointing this out. We have added these contents to the revised manuscript.

Line 41: Abstract is clear but the last sentence should be in the discussion not in the abstract “In conclusion, the selective pressure of quinolones caused CC4821 to converge evolution, which is more invasive and spreading”

Re: Thanks. We have modified this sentence according to your suggestion.

Line 251-252. It appears there are 2 sentences here. Please revise the statement.

Re: Thanks a lot. We have modified this sentence.

Line 267: should be 4 cases,...

Re: Yes, You're right. We have corrected it.

Line 271-272: when patients clinically suspected of meningococcal meningitis but the culture is negative. mNGS can be used to confirmed the etiological diagnosis as soon as possible.

Suggest the author should indicate “how long each test takes to deliver the result”.

Re: Thanks. We have added the time limit for mNGS in the revised manuscript.

Line 298-300: need the reference for this statement “...fluoroquinolone selective pressure is the main reason for the outbreak of CC4821 in China”

Re: As suggested, we have added one reference to support this idea.

Line 302: should it be quinolone resistant “gene” from three commensal....

Re: Thanks to you for your good comments, we have added it.

Line 306: Add a space between ...France (2009-2011) [39].

Re: Thanks. We have added it.

Line 336-338 and Supplemental Dataset 2: The spreadsheet shows that Probability value of Shanghai-China is 0.20199, while Beijing_China has only 0.06215. However in the file legend, it said “Beijing might be the root of CC4821 with over 20% probability”. A little confusing here, please clarify.

Re: We enlarged the sample size and removed Supplemental Dataset 2.

REVIEWERS' COMMENTS:

Reviewer #1 (Remarks to the Author):

Thank you very much for addressing my comments, I am pleased with the results and recommend that the manuscript is published.

Reviewer #2 (Remarks to the Author):

Recommend to accept now. Happy with revisions